Artificial intelligence-driven insights into Arab media’s sustainable development goals coverage

Alsuhaibani Mohammed 1
Gaanoun Kamel 2
http://orcid.org/0000-0002-5323-6661 Qamar Ali Mustafa 1 al.khan@qu.edu.sa
1 Department of Computer Science, College of Computer, Qassim University , Buraydah , Saudi Arabia
2 INSEA , Rabat , Morocco
Kong Xiangjie
Electronic publication date: 2025 Aug 26
Publication date: 2025
Volume: 11
Electronic Location ID: e3071
Received 2025 Jan 20; Accepted 2025 Jul 3
Copyright: © 2025 Alsuhaibani et al.
Copyright year: 2025
Copyright holder: Alsuhaibani et al.
License: This is an open access article distributed under the terms of the Creative Commons Attribution License, which permits unrestricted use, distribution, reproduction and adaptation in any medium and for any purpose provided that it is properly attributed. For attribution, the original author(s), title, publication source (PeerJ Computer Science) and either DOI or URL of the article must be cited.
License URL: https://creativecommons.org/licenses/by/4.0/

Keywords: Sustainable development goals, SDGs, Arab media, Arabic NLP, Machine learning, Artificial intelligence, Deep learning, Transformer models

Funding: Qassim University Deanship of Scientific Research 2023-SDG-1-BSRC36236 This work was funded by Qassim University, represented by the Deanship of Scientific Research, under the number (2023-SDG-1-BSRC36236) during the academic year 1445 AH/2023 AD. The funders had no role in study design, data collection and analysis, decision to publish, or preparation of the manuscript.

==============================
This study examines how Arab media have engaged with the United Nations Sustainable Development Goals (SDGs) over the past decade and evaluates the alignment between media coverage and official government priorities. The research addresses the lack of large-scale, Arabic-focused analyses in SDG discourse, which is often dominated by English-language studies. We collected and processed a unique dataset of over 1.2 million Arabic news articles from ten countries between 2010 and 2024. Using a combination of data augmentation, deep learning (specifically, Transformer-based models), and large language models (LLMs), we trained classifiers to detect references to the SDGs and categorize articles by specific SDGs. The results reveal regional patterns in SDG coverage, with North African countries focusing more on governance-related goals, while Gulf countries emphasize economic and environmental themes. Our findings reveal a general alignment between media discourse and official SDG priorities, with notable exceptions. This study is the first to combine artificial intelligence (AI) methods and Arabic media at this scale for SDG analysis, offering new tools and insights for policymakers, media professionals, and development stakeholders.

Introduction

The Sustainable Development Goals (SDGs) constitute a crucial global framework for addressing humanity’s most pressing challenges, ranging from poverty alleviation to climate action. Governments, businesses, and civic society have turned their primary attention toward these goals (Maguire et al., 2024; Stafford-Smith et al., 2017). The media’s role in disseminating knowledge about the SDGs becomes increasingly important as these objectives permeate global dialogue. Media coverage not only tells the people but also reflects and shapes public and governmental policies. In the Arab world, where media are as varied as the civilizations and societies they depict, the coverage of SDGs offers unique challenges and opportunities (Krzymowski, 2020; Tahat et al., 2024).

Arab media serve as a critical channel between the public and policy-making entities. However, limited research has examined how well these media effectively cover the SDGs, particularly within the Arabic-speaking world, which remains underrepresented in global media analysis due to a prevailing focus on English-language data sources. Existing studies on Arab media mainly analyze smaller datasets (Marzouki et al., 2021), short time periods (e.g., 6 months as in Mahdi & Ghntab (2019) and 1 year as covered by Tahat et al. (2024)), or only a few countries (one in case of Singh et al. (2022) and Tahat et al. (2024)). Furthermore, most research works rely on manual analysis or basic tools not designed for Arabic, making large-scale studies difficult.

Furthermore, there is a notable gap in the literature regarding automated and scalable approaches to assess how well media narratives reflect or diverge from governmental SDG agendas in Arabic-speaking countries. These gaps hinder the ability to comprehensively evaluate public discourse and its alignment with national and international development frameworks.

To address these gaps, the primary objective of this study is to explore the extent and nature of SDG coverage in Arabic newspapers from ten countries using large-scale, automated content analysis. This work is novel in both its linguistic scope and technical methodology. By doing so, we aim to assess whether Arab media shares the same priorities as governments concerning sustainable development. To achieve this goal, we propose a comprehensive and varied dataset, the largest of its kind, of Arab news articles, with a particular emphasis on detecting mentions of the SDG. Furthermore, a significant contribution of this study is the development of advanced methods for analyzing Arabic media content. To address the challenges posed by the intricate and varied nature of the Arabic language, we have undertaken the task of translating and augmenting an existing dataset, known as the OpenSDG (OSDG) (OSDG, UNDP IICPSD SDG AI Lab & PPMI, 2024). This dataset initially comprised English texts tagged with labels related to the SDGs. The augmented dataset was utilized for training machine learning (ML) models. In particular, we employed deep learning (DL) models, including transformer language models (TLMs) (Brown, 2020; Devlin, 2018; Vaswani, 2017) and large language models (LLMs), such as GPT-4, which have recently gained considerable importance.

The class of models known as Transformers has significantly transformed the domain of natural language processing (NLP) by enabling machines to comprehend and produce human language. In this study, we adapt these Arabic language models to create the first specialized tools designed to identify references to the SDGs within Arabic text. The tools include binary classifiers that determine whether an article references the SDGs and multi-class classifiers that identify specific SDGs mentioned in the text. The analysis provided by these models enables us to undertake a comprehensive investigation of the engagement of Arabic media with the SDGs. To assess the congruence between public discourse and governmental priorities, we compare the frequency and focus of SDG topics in media articles, identified using our trained classification models, with country-specific indicators and official reports published by national governments and international bodies. This comparative approach enables us to examine the presence of alignment and the nature of any divergences, providing insights into the extent to which the media’s representation aligns with or deviates from official sustainable development strategies. Furthermore, our work enables a comparison among several Arab countries, uncovering regional differences and patterns in the coverage of SDGs.

Through this research, we aim to contribute to a deeper understanding of the role of Arab media in promoting sustainable development. Our analysis highlights the correlation, or lack thereof, between media coverage and official SDG priorities, providing valuable insights for policymakers, media professionals, and researchers interested in the intersection of media, public policy, and sustainable development.

The main contributions of this article are as follows: Curation of the largest dataset of Arab news articles from 10 countries, spanning from 2010 to 2024, with a focus on SDG-related content

Translation and augmentation of the OSDG dataset to include Arabic texts, enabling the analysis of SDGs in Arabic media

Detection of SDG-related articles in Arabic newspapers using advanced ML techniques

Identification of specific SDGs covered in Arab news articles, offering a detailed view of media engagement with each goal

Development and publication of two Transformer models for Arabic text, capable of performing both binary and multi-class classification of SDG references

Analysis of the alignment between Arab media’s SDG coverage and official statistics, providing insights into the match between public discourse and governmental priorities

Our findings are discussed in the context of previous literature on SDG media representation, highlighting both shared trends and region-specific deviations. Our analysis reveals distinct regional patterns in SDG coverage: North African media emphasize governance (SDG 16) and social issues, while Gulf countries focus on economic growth (SDG 8) and environmental sustainability. Significant events, such as COVID-19 and the Arab Spring, significantly influenced coverage trends, with health (SDG 3) and justice-related content showing the strongest correlations. More importantly, we identified discrepancies between media priorities and official United Nations (UN) progress reports, particularly in the areas of poverty and climate change coverage. The study also demonstrates the effectiveness of our Arabic-language artificial intelligence (AI) models, achieving 87% accuracy in SDG classification.

The rest of the article is organized as follows: We first provide an overview of the existing literature and the methodology employed in this study. We then describe our data collection and augmentation approach, followed by the development of models for classifying SDG-related content. Finally, we present the results and discuss their implications for understanding the alignment between Arab media coverage and the official SDG priorities.

Literature review

The UN presented the 2030 Agenda for Sustainable Development in 2015, which was adopted by all UN member states that same year. The agenda includes 17 SDGs, ranging from ending poverty and other deprivations to good health, education, and climate action. The UN Secretary-General submits a progress report every year in this regard. The role of media in steering public attention and framing policy agendas about these SDGs has increasingly garnered scholarly attention. However, a predominant focus on English-centric analyses has inadvertently marginalized non-English contexts, particularly those about Arabic-speaking regions. This literature review aims to explore the intersection of media influence, machine learning (ML) methodologies, and the scope of the SDGs within the context of the Arabic-language media landscape.

Arabic news datasets

Many researchers have contributed to the development of diverse Arabic news datasets. Nagoudi et al. (2020) developed AraNews, a publicly available large-scale and multi-country parts-of-speech (POS)-tagged dataset containing data from 50 newspapers and 17 countries (15 Arab along with USA, and UK). It contains more than 5.1 million articles. However, our focus is exclusively on articles obtained from news websites in Arab countries. Furthermore, AraNews does not include data from Hespress (Morocco), Tunisie-telegraph (Tunisia), Sabq (Saudi Arabia), Kuwait News (Kuwait), and Aletihad (UAE), all of which have been considered in this research. Some of the Arab countries are not covered at all. Furthermore, no timeframe is mentioned in the data.

Altamimi & Alayba (2023) created a Modern Standard Arabic (MSA) dataset, named Arabic News Article Dataset (ANAD), containing more than 0.5 million news articles. The articles were obtained from 12 websites, and the data were retrieved over a 1-year period (2021). The data included ten categories, such as sports, economy, politics, entertainment, cars, health, and art, showing the diversity in Arab news content. Sports contained the maximum number of articles ( 34.7%), followed by politics ( 18.5%) and economy ( 13.4%). The dataset is geographically limited, and most articles are from Saudi Arabia. Furthermore, no longitudinal analysis could be performed because the entire dataset is from a single year.

Similarly, Einea, Elnagar & Al Debsi (2019) developed the Single-label Arabic News Articles Dataset (SANAD) for text classification. However, the data were raw and obtained from only three news websites. The number of articles was around 194 K. The data contained seven categories: culture, finance, medicine, politics, religion, sports, and technology. The most number of articles belonged to the finance category.

Honti et al. (2021) present a dataset specifically built for news-based monitoring of the SDGs. The dataset is obtained by running structured queries of the Global Database of Events, Language and Tone (GDELT) database. GDELT utilizes NLP, data mining, and DL to extract and monitor global news in over 100 languages and from countries worldwide. The data stretches back to 1979 and is updated every 15 min. The researchers show the data related to SDG 6–Clean Water and Sanitation as an example. However, any of the 17 SDGs could be analyzed in detail. The total number of articles is more than 5.6 million from 226 countries.

Media, SDG engagement, and predictive modeling for SDG analysis

Marzouki et al. (2021) analyzed the tweets related to SDG 11 (covering sustainable cities and communities). However, only tweets written in English are considered. The tweets appearing between September 2015 and December 2020 (a period of approximately 5 years) are considered. After preprocessing, more than 0.7 million tweets remained for further analysis, of which only slightly more than 31,000 referred to SDG 11. The most tweets appeared in 2019. The researchers also found that the importance of hashtags related to AI, the Internet of Things (IoT), and innovation, when discussing cities, gradually increases with each passing year. A drawback of this research is that no semantic data analysis was conducted.

López-Carrión & Martí-Sánchez (2024) performed a longitudinal study on the coverage of SDGs in the Spanish digital press. They analyzed 8 years of data (2015–2022) belonging to ten digital newspapers with the highest readership in Spain. The number of relevant news articles increased from 773 in 2015 to 8,509 in 2022, representing a growth of more than 1000%. Just like Marzouki et al. (2021), the maximum growth in the number of news articles was seen in 2019. The news content was clustered into four thematic clusters, including global issues (representing 26.5% of the articles), UN ( 27.3%), Institutions ( 24.8%), and SDG Objectives and Targets ( 21.4%), using the bisecting K-means algorithm.

Chenary, Pirian Kalat & Sharifi (2024) tried to forecast the SDG score of different regions by 2030. The data was obtained from the SDG progress report published by Dublin University Press in 2023. The data contain worldwide SDG scores, 17 goals, and 169 targets from 2020 to 2022. They employed two ML techniques: AutoRegressive Integrated Moving Average with eXogenous inputs (ARIMAX) and linear regression (LR). The highest scores were achieved by the Organisation for Economic Co-operation and Development (OECD) countries, with a mean score of 75.05 and the lowest standard deviation of 1.78, followed by Eastern Europe and Central Asia, with a mean score of 68.42. The Middle East and North Africa, with a mean score of 63.88, are among the weakest regions in attaining the SDG score. According to the predictions made by ARIMAX, OECD countries are expected to attain a score of 79.86 by 2030, whereas Eastern Europe and Central Asia are projected to achieve a score of 73.88. However, the Middle East and North Africa would only be able to get a score of 68.62. Fairly consistent results were exhibited by LR.

Czvetko et al. (2021) analyzed the SDGs based only on the news articles. The work primarily involved network analysis, and a news-centered approach was used to model the interconnections between countries regarding sustainable development actions. The data was retrieved from the GDELT project. More than 60 million news stories from 2019 were considered. The results showed that the share of the sustainability news for Egypt, Libya, and Saudi Arabia is just 1.62–1.75%. In the case of Oman, another Arab country, the share increased to 1.84–2%. However, for Morocco, the share decreased to 1.52–1.62%. Similarly, for Algeria, the share appeared as 1.42–1.52%. The maximum share of 2–2.62% was observed for China and the central African countries, such as the Democratic Republic of the Congo, Tanzania, and Zambia. The researchers also studied SDG 13—The Climate Change. Algeria, Iraq, and Libya were among the countries with the lowest proportion of news related to climate change (0–0.0112%). Morocco and Saudi Arabia showed a slightly higher share of 0.0112–0.0177%. However, Egypt appeared in the second-best category (0.0465–0.0703%). The USA emerged as the most active player. They only considered the data from 2019, and therefore, no longitudinal analysis was performed. Lastly, no AI was employed in this research.

Alharbi (2023) also worked on predicting SDG accomplishment. He employed hierarchical classification based on time series modeling to indicate which of the 38 geographical entities would meet their SDG. More precisely, he used an encoder-decoder long-short term memory (LSTM) recurrent neural network (RNN)-based model for multivariate forecasting. A taxonomy was developed based on the raw SDG data provided by the United Nations Department of Economic and Social Affairs’ Statistics Division. The results confirmed that individual SDGs are not independent of one another.

Four national newspapers were considered from January 2016 to June 2017. The coverage placement, which could be on the front page, inside pages, center spread, or the back page, was also studied. Most of the discussion on SDGs appeared on the inside pages. Studying the direction of the reportage, they found that most (more than 60%) of the articles reported issues on SDGs in Nigeria positively. We observed that only the frequency of newspaper articles discussing SDGs was analyzed, and no content analysis was performed.

Singh et al. (2022) discussed the impact of the SDGs in Saudi Arabia. The researchers primarily focused on economic growth and discussed the role of education in achieving these goals under Saudi Vision 2030. Of the 17 SDGs, eight (1–5, 8, 9, and 13) were analyzed using data spanning 30 years (1990–2020) and obtained from diverse sources, including Saudi Arabia General Authority for Statistics, International Labor Organization (ILO), Food and Agriculture Organization (FAO), UN Sustainable Development Report, World Health Organization (WHO), World Bank, and UN. Their results showed that education and training (SDG 4: educational quality and lifelong learning) have the highest contribution to the GDP growth rate. While the study is economic in nature, it highlights an essential point about the media’s potential role: disseminating such findings through media can shape public understanding and policy focus on key SDGs.

Mahdi & Ghntab (2019) studied the SDGs in the Arab Press. They selected three newspapers, one each from Iraq (Al-Sabah), Algeria (Ech-chaab), and UAE (Aletihad). The articles were obtained in a 6-month window (from Jan 1 to June 30, 2018). The research showed that only three SDGs (8—economic growth and decent work, 11—sustainable local cities and settlements, and 16—peace, justice and institutions) were prominently discussed in the Arab press. This analysis highlights a selective engagement with the SDGs in the Arab press, suggesting potential areas where media focus could be expanded. Nevertheless, the data was limited, and no longitudinal analysis could be performed.

In another exciting study, Tahat et al. (2024) discussed the coverage of the SDGs in various newspapers in the UAE. They considered both Arabic and English-language newspapers. Aletihad was selected from Arabic newspapers, just like our research, whereas Gulf News was chosen from English newspapers. The data were gathered from Nov 7, 2021, to Nov 3, 2022. The articles related to SDGs peaked in January 2022 ( 13% of all articles). The results showed that 88% of the articles positively discussed sustainable issues. Interestingly, environmental issues are the most frequently discussed SDG. However, the lack of NLP techniques in their analysis leaves room for advanced computational approaches to enhance the understanding of media discourse.

From the review of the current research, several gaps were observed in how media coverage of the SDGs is made. The gaps are more evident in the case of Arabic countries. Most of the research focuses on English-speaking or non-Arab countries. Even when Arabic-speaking contexts are covered, the studies are mostly limited to shorter periods or a few countries, as observed in the 6 months covered by Mahdi & Ghntab (2019) or the 1 year covered by Tahat et al. (2024). Furthermore, most studies employ simple or even manual methods for content analysis. For example, Czvetko et al. (2021) conducted network analysis on SDG-related news but did not use advanced AI or deep learning techniques that could offer more profound insights. Similarly, studies such as Marzouki et al. (2021) and Tahat et al. (2024) did not employ advanced NLP or ML techniques, which limits the understanding of the language and sentiment surrounding SDGs in the media.

Furthermore, very few studies provide an in-depth, long-term analysis of how SDG coverage in the media changes over time. While López-Carrión & Martí-Sánchez (2024) conducted a thorough analysis in Spain, similar studies are missing for Arabic media. This would help us understand how discussions evolve in response to regional and global events. Some datasets, such as AraNews by Nagoudi et al. (2020) and OSIAN by Zeroual et al. (2019), are helpful but have issues, including limited geographic coverage, outdated content, or a lack of clarity regarding the period they cover.

This study aims to address these gaps by developing the largest dataset of SDG-related Arab news, covering 10 countries over the last 14 years. Moreover, we develop DL techniques specifically for Arabic text for both binary and multi-class classification. By doing so, we hope to provide a more detailed, comprehensive, and long-term analysis of how the SDGs are covered in Arabic-speaking media, contributing important new insights to this field. Table 1 compares our approach and previous research.

Table 1 Comparative analysis of our study vs. previous research.

Aspect	Previous research	Current study	
Language focus	Primarily English-centric analyses (e.g., Marzouki et al., 2021; Czvetko et al., 2021)	Focuses exclusively on Arabic language media, addressing a critical gap.	
Geographic coverage	Limited to a few Arab countries or short timeframes (e.g., Mahdi & Ghntab, 2019; Tahat et al., 2024).	Covers 10 Arab countries (2010–2024), enabling regional and longitudinal analysis.	
Dataset size	Smaller datasets (e.g., 194K articles in SANAD; 0.5 M in ANAD).	More than 1.2 million articles, the largest Arabic SDG-focused dataset to date.	
Methodology	Manual analysis or simple NLP techniques (e.g., Tahat et al., 2024; Czvetko et al., 2021).	Uses Transformer models (ArBERTv2) and LLMs (GPT-4) for automated, scalable analysis.	
SDG classification	Rarely addressed; limited to binary classification (e.g., Honti et al., 2021).	Hierarchical approach: Binary + multi-class SDG classification (16 goals).	
Alignment with policy	Minimal comparison with official SDG priorities (e.g., López-Carrión & Martí-Sánchez, 2024).	Explicitly analyzes media-government alignment using UN SDG Index reports.	
Data diversity	Focus on news or tweets (e.g., Einea, Elnagar & Al Debsi, 2019; Marzouki et al., 2021).	Combines real news articles with synthetic data for robustness.	
Temporal scope	Short-term (e.g., 6 months in Mahdi & Ghntab (2019)).	14-year span (2010–2024), capturing pre-/post-SDG adoption and crises	

Methodology overview

The primary objective of this research is to analyze Arabic news articles and assess the extent to which they address topics related to the UN SDGs. The methodology is divided into several key phases: data collection, preprocessing, model development, classification, and insight generation.

Data collection: The study utilizes two primary data sources: training data and inference data. The training data is sourced from the OpenSDG (OSDG) dataset (OSDG, UNDP IICPSD SDG AI Lab & PPMI, 2024), which consists of English texts annotated with corresponding SDG labels. Synthetic data is generated using ChatGPT to augment this dataset. This synthetic data supplements the OSDG dataset with additional SDG-related and non-SDG articles to facilitate binary classification. It also provides text that resembles news articles to address the OSDG dataset’s focus on official documents and reports. Inference data comprises articles scraped from well-known Arabic news websites. These articles are selected based on specific criteria to ensure their relevance and quality. The time window for data collection spans from 2010 to 2024, providing broad temporal coverage to capture trends over time.

Data preprocessing: In the data preprocessing phase, the OSDG dataset is translated from English to Arabic to ensure consistency in the language of the training data. Both the translated and synthetic datasets are preprocessed to remove noise and ensure suitability for model training.

Model development: Following data preparation, a hierarchical model was developed and trained. The first step involved a binary classifier to distinguish between SDG-related and non-SDG-related items. Subsequently, a multi-classifier was applied to the positively classified articles to determine the specific SDGs covered in each. These models are fine-tuned using the preprocessed training data to optimize their performance.

Classification: The trained models classify the items in the inference dataset. The binary classification approach decides if an item is relevant to the Sustainable Development Goals, whereas the multi-class model assigns a specific SDG category to each relevant item. This classification process thoroughly examines SDG-related material in Arab press stories.

Insight generation: The last element of the process entails developing conclusions from the classification findings. Classified articles are analyzed to explore regional comparisons and the overall evolution of presence and interest in the SDGs in the Arab media. This data is crucial for understanding the various levels of commitment to the SDGs in Arab countries, providing valuable insights for policymakers, scholars, and organizations working on sustainable development.

The subsequent sections will provide a detailed discussion of each of the aforementioned methodological steps.

Data collection

This section describes the method used to identify relevant news sources, extract data, translate content, and generate synthetic data for model training and evaluation. This study utilized a dataset comprising authentic Arabic newspaper articles and synthetically generated articles. Two separate datasets were created: a training dataset for developing and refining classification models and an inference dataset for applying the models and generating predictions for news analysis.

Arabic news article collection (inference data)

Hespress (Morocco), Elkhabar (Algeria), Tunisie-telegraph (Tunisia), Youm7 (Egypt), Sahara Media and AMI (Mauritania), Sabq (Saudi Arabia), Kuwait News (Kuwait), Marsalqatar (Qatar), Aletihad (UAE), and Shabiba (Oman).

These sources were selected based on specific criteria: the websites needed to be country-specific rather than international news outlets like Al Jazeera or BBC Arabic. This approach ensures that the data more accurately reflects local perspectives. Additionally, the chosen sources were required to have at least three years of archived data, with five years being preferred, and must be accessible (ideally with a sitemap). The goal was to cover a diverse range of Arab regions. Articles written in Modern Standard Arabic (MSA) were given priority to ensure consistency and clarity in the dataset.

For websites equipped with sitemaps—a structured file that provides information about the pages on a site—Octoparse, a web scraping tool, was utilized to extract article URLs and their content directly. A sitemap helps to efficiently navigate and extract data from a website by listing all the pages available for crawling. Python’s Selenium library was used to gather article URLs for websites without sitemaps. These URLs were subsequently processed through Octoparse for content extraction. This dual-method approach streamlined the data collection while adhering to the constraints specified in each website’s robots.txt file, which governs how and what data can be scraped from the site.

The scraping targeted four specific fields: the article’s date, the category (when applicable), the title, and the content. Among these fields, the date and the content were the minimum required; the date is essential for longitudinal analysis, while the content is necessary for evaluation and classification. A total of 1,945,783 articles were initially collected before any preprocessing steps were taken. The web scraping process required a cumulative duration of 121,000 h.

Data preprocessing and validation

The collected data underwent a series of preprocessing steps to ensure quality and consistency. The validation process involved language verification, relevance checks, and anomaly detection. Several steps were applied to ensure the relevance and quality of articles for inclusion in the final dataset: Removal of articles with empty paragraphs

Removal of special characters: \n and \t

Removal of articles written in French (Elkhabar Category “Articles en Français”) (انضم إلى قناتنا على واتساب “nDm ‘iilA qnAtnA ‘Ala wAtsAb’”)

Removal of webpage errors included in the articles (Youm7 and Sabq)

Removal of articles with paragraph lengths less than 20 words

Deletion of irrelevant categories of articles: Videos and images or caricature articles, Advertisement articles, and Horoscope.

Specific regular expression (regex) functions were then developed for each website to extract the article year from the date field. Articles lacking a date field were subsequently deleted.

After a preliminary analysis, two websites required additional attention. The data from Hespress spanned from 2007 to 2024. However, this website only became a news media outlet in 2010 when it began hiring professional journalists; before that, it was a simple blog with articles written by bloggers. Therefore, it was decided to disregard Hespress data prior to 2010. The AMI website was identified as not complying with the study objectives, as it is an official website that publishes the actions of the Mauritanian government. Consequently, it was decided to exclude this website from the study.

After applying these criteria and removing duplicates, the final dataset comprised 1,274,266 articles. This substantial corpus provided a robust foundation for the subsequent phases of model training and analysis, enabling a comprehensive exploration of the presence of SDGs in Arabic news media across different regions and over time. The main statistics about the inference data are presented in Table 2 and Fig. 1.

Table 2 Inference dataset main statistics.

Website	Nb of articles	Time span	Article average length	
Aletihad	42,022	2010–2024	441	
Elkhabar	118,733	2015–2024	123	
Hespress	332,926	2010–2024	397	
KuweitNews	30,106	2021–2024	222	
Marsalqatar	49,843	2021–2024	211	
Sabq	481,676	2014–2024	161	
SaharaMedia	17,124	2017–2024	203	
Shabiba	39,274	2016–2024	301	
TunisieTelegraph	39,869	2015–2024	184	
Youm7	122,693	2017–2024	223	
Total	1,274,266	2010–2024	238	

Figure 1 Number of articles per website.

OpenSDG dataset

The OpenSDG (OSDG) dataset (OSDG, UNDP IICPSD SDG AI Lab & PPMI, 2024) serves as the foundation for the training data in this work. This dataset comprises 42,635 text excerpts from official documents, reports, and research studies, each labeled with the corresponding relevant SDG. These extracts were validated by volunteers using the OSDG community platform, ensuring a certain level of quality control and consensus during the annotation process. OSDG is an open-source project that organizes research publications and other text resources according to the United Nations Sustainable Development Goals. The excerpts were initially labeled using an ML model, and the accuracy of these labels was then improved through a crowdsourcing effort. Volunteers from the OSDG Community Platform reviewed and validated the SDG classifications, ensuring high quality and consensus in the labeling process.

An important aspect of the OSDG dataset is the agreement score associated with each labeled excerpt. The agreement score represents the consensus among the community reviewers regarding the SDG classification of a particular text. It is calculated as the percentage of reviewers who agreed with the assigned SDG label. For instance, an agreement score of 0.8 indicates that 80% of the reviewers agreed with the given SDG classification for that excerpt. The agreement scores in the OSDG dataset range from 0 to 1, with higher scores indicating stronger consensus. This metric provides valuable information about the reliability of the SDG labels and can be used to filter the dataset based on the desired level of annotation confidence.

To adapt the OSDG dataset for the study’s context, all 42,635 text excerpts were translated from English to Arabic using the Google Cloud Translation API. This translation process enabled the creation of a parallel dataset that retained the original SDG annotations while providing Arabic language content for model training. The fully translated dataset has been made publicly available to facilitate and support future research efforts.

In this study, only excerpts with an agreement score of 0.7 or higher were utilized, ensuring a high level of reliability in the training data. This filtering process resulted in a subset of 22,030 records, which formed the basis of the training dataset.

The dataset covers SDGs from 1 to 16, with varying numbers of examples for each goal. This distribution reflects the real-world emphasis and research focus on various SDGs. For instance, SDGs related to health, education, and economic growth tend to have more examples than those related to specific environmental issues. SDG 17 is not represented in the OSDG dataset because it focuses on partnerships and implementation mechanisms, which are difficult to capture in text. For this reason, it is also excluded from this study. The distribution of SDG labels in the dataset is depicted in Fig. 2.

Figure 2 SDGs distribution in OSDG dataset.

The OSDG excerpts are relatively concise, averaging 94 words and approximately five sentences. The translated versions are similarly brief, averaging 88 words. This brevity is attributed to the OSDG’s focus on excerpts containing specific SDG-related content rather than entire documents. The use of the OSDG dataset provided several advantages for this study: Comprehensive coverage of all SDGs (except SDG 17, due to the challenges associated with its analysis as stated by the dataset creators (Pukelis et al., 2022))

High-quality, validated annotations with associated agreement scores

Open-source nature, allowing for reproducibility and further research

However, there are notable limitations that must be addressed: Limited size of dataset subset: The subset of the OSDG dataset used in this study is relatively small, which may restrict the generalizability of the findings.

Focus on official documents and scientific papers: The OSDG dataset primarily includes official documents and scientific papers. While it also covers news articles, these are not the main focus, potentially limiting the breadth of perspectives in the analysis.

Exclusion of non-SDG texts: The dataset only includes texts related to the SDGs, excluding those not aligned with these goals. This exclusion may result in a narrow view, missing broader contextual information.

Language limitation: The OSDG dataset is presented in English, while this study explicitly targets Arabic articles, necessitating translation and potentially introducing language-related biases.

Data augmentation with ChatGPT

As the translated OSDG dataset mainly consisted of excerpts from formal documents, it had limitations in terms of size and diversity. In order to improve the training dataset, synthetic data was generated by using the ChatGPT API, an advanced language model known for its ability to produce human-like text. To accomplish this, we created detailed prompts for producing both SDG-related and non-SDG-related news articles, modeled after the style and length of real-world news articles.

Generating only synthetic non-SDG texts while relying solely on the OSDG dataset for SDG-related content could introduce potential bias during model training, as the model might differentiate between the two types based purely on the synthetic vs. non-synthetic nature of the data. This risk was mitigated by creating synthetic SDG articles and integrating them with the OSDG data to ensure that the model learns from a more balanced and realistic dataset. In this way, consistency is maintained, and biases are less likely to develop based on the data source.

The specific prompts used in the synthetic data generation process were designed to produce content similar to news articles, emphasizing the goals of the SDGs through indirect discussion rather than explicit mention. This method enabled us to ensure that the synthetic articles accurately reflected the actual news content as closely as possible. The following section provides information on the prompt used.

Prompt design

The primary objective of the prompt construction was to produce synthetic news articles that accurately reflected real-world content in terms of style and length, while clearly distinguishing between material related to the SDGs and that which was not. It was imperative that the texts created were entirely in Arabic, with no English terms, and that they did not directly mention specific SDG names or numbers. This design ensured that the distinction between articles related to the Sustainable Development Goals and those that were not was based on concrete content rather than simple keyword matching.

An iterative process of testing and refinement was implemented to create a prompt that met these criteria. Initial experiments with GPT-3.5 were unsatisfactory, as the generated texts often did not adhere to the required news article style and sometimes contained English words despite clear instructions to the contrary. GPT-4 was then employed, resulting in considerably improved performance.

ChatGPT was assigned the role of expert on SDGs and Arabic news articles, using its experience to distinguish SDG-related content from non-SDG content. It was able to create articles with a better understanding of the differences between the two categories.

The prompt was designed to include two primary commands: one for generating articles related to SDGs and another for non-SDG articles. When generating SDG-related content, the command would specify which SDG to focus on, guiding the generation process accordingly. Conversely, the non-SDG command directed the model to create content unrelated to SDGs, using the same overall prompt framework but with a different focus.

One of the key challenges encountered with GPT-4 was controlling the length of the generated articles to match the scraped articles’ average length. Instead of specifying a word count, which proved challenging to manage, the prompt was adjusted to create articles based on the number of paragraphs. Through testing, it was determined that four paragraphs typically resulted in article lengths comparable to those found in the inference dataset.

A key challenge was to ensure that the prompt clearly distinguished between SDG-related and non-SDG content. To tackle this, the prompt incorporated a distinction based on ChatGPT’s understanding of the difference. This understanding stemmed from a separate prompt where ChatGPT was asked to explain how it differentiates between SDG and non-SDG articles across several dimensions, including theme, content, language, terminology, purpose, and perspective. The insights from this process were then integrated into the final prompt to enhance the model’s ability to generate content aligned with SDG objectives.

The prompt instructed GPT-4 to generate articles following clear guidelines. Regarding SDG-related content, the articles focused on initiatives, progress, challenges, or achievements related to specific SDGs, emphasizing themes such as sustainability, social progress, environmental protection, and economic development. Rather than directly referencing the SDGs, the articles used contextual cues, enabling the model to recognize relevant material without explicit mentions.

In contrast, non-SDG-related articles were designed to cover a broad range of topics-such as politics, entertainment, science, technology, sports, or human interest stories-without directly focusing on sustainable development. These articles aimed to inform, entertain, or engage readers on subjects unrelated to the SDGs without using sustainability-specific terminology.

By incorporating these refined criteria into the prompt, the generated articles closely resembled the content of real news stories. This method enabled the clear distinction between SDG-related material and non-SDG-related material, while preserving the style and length typical of news articles. The complete prompt is available in Appendix A “ChatGPT Prompt”.

While GPT-4 demonstrated strong capabilities in Arabic text generation, it is primarily trained on English and Western-oriented content. This may introduce subtle biases in how SDG themes are framed or contextualized in the generated articles. Although efforts were made to ensure cultural relevance, we acknowledge that the model’s global training data may affect the authenticity of synthetic Arabic texts. This limitation is mitigated by combining synthetic and real-world texts during training; however, it remains important to consider when interpreting the results.

The parameters utilized for the generation process are detailed in Table 3.

Table 3 Parameters used in the generation process.

Parameter	Value	
Model	GPT-4	
Temperature	1	
Maximum length	2,000	
Top P	1	
Frequency penalty	0	
Presence penalty	0	

Synthetic articles

ChatGPT was used to generate 7,406 synthetic articles, including 3,307 related to the SDGs and 4,099 unrelated. The entire dataset comprises around 22 million tokens. Manual inspection of a sample of the generated articles confirmed that they adhered to expected standards in both style and semantic relevance.

The generation process followed a targeted strategy. For each SDG label in the OSDG dataset, 15% more articles were created, preserving the original label proportions and enabling cost-effective dataset expansion. This resulted in 3,307 additional SDG-related articles with label distributions aligned to those of the original dataset.

Given the natural scarcity of non-SDG-related texts in labeled datasets, 4,099 unrelated articles were also generated. Although limited by cost and making up 14% of the overall dataset, this imbalance was mitigated during training through undersampling.

The final dataset consisted of 22,030 OSDG articles and 7,406 synthetic ones, totaling 29,436. This corresponds to a 15% augmentation for SDG data and a 34% overall increase when non-SDG-related content is included. The synthetic dataset has been made publicly available to support future research.

Model development

The study employed a hierarchical classification approach, utilizing the ArBERTv2 model (Abdul-Mageed, Elmadany & Nagoudi, 2021) as the foundation for binary and multi-class classification tasks. ArBERTv2 is a deep bidirectional transformer encoder based on the BERT-base architecture with 12 layers, 768 hidden units, and 12 attention heads. It was pre-trained on more than 200 GB of Modern Standard Arabic (MSA) texts, including news articles, making it particularly suitable for this study’s focus on Arabic media.

Binary classification model

The binary classification model, serving as the first layer of the hierarchical approach, aimed to distinguish between articles related to SDGs and those unrelated. To implement this, a classification head with a sigmoid activation function was added to ArBERTv2. All transformer layers were fine-tuned during the training process.

The dataset was split into training, validation, and test sets with proportions of 75%,12.5%, and 12.5%, respectively. Due to the class imbalance, where non-SDG articles vastly outnumbered SDG ones, undersampling was applied to balance the training set. The SDG class was adjusted to include 50% real (from OSDG) and 50% synthetic examples, preserving the original distribution of SDG labels through stratified sampling.

The model was trained using the AdamW optimizer, with a learning rate of 5e-5, a batch size of 16, and for three epochs. The training was repeated with three different random seeds to ensure robustness. The model achieved an average macro-F1 score of 98% on the test set. A manual review was conducted to further validate its practical performance by comparing predictions with human annotations on randomly selected samples, yielding an average accuracy of 85%.

Multi-class classification model

The multi-class model, forming the second layer of the classification pipeline, was designed to identify the specific SDG associated with articles previously labeled as SDG-related. A multi-class output layer with softmax activation was added on top of ArBERTv2, and the entire model was fine-tuned during training.

The input data consisted of SDG-labeled texts from the augmented OSDG dataset. Stratified splitting ensured that both SDG label distributions and text sources (OSDG vs. synthetic) were preserved across the training (60%), validation (20%), and test (20%) sets.

Training followed the same setup as the binary model: AdamW optimizer, learning rate of 5e-5, batch size of 16, and 5 epochs, with a fixed seed of 42 for reproducibility. The model achieved an average macro-F1 score of 87% on the test set. A manual evaluation, following the same procedure as in the binary case, yielded an average accuracy of 79%, indicating reliable identification of specific SDGs.

This hierarchical setup allowed efficient use of available data. The initial binary classifier, trained on a balanced dataset, simplified the downstream task by reducing class noise. The subsequent multi-class classifier then leveraged the full SDG-related subset for fine-grained label prediction.

Both models are publicly available on the Hugging Face model hub.

Results and discussion

This section presents the results, including the coverage of SDGs by various media outlets, SDG priorities in the Arab news, and the ensuing discussion.

Overall SDG coverage

This analysis presents the data as the proportion of SDG-related articles among the total number of articles published by each website. This approach addresses and alleviates the issue of discrepancies in the number of articles scraped from each site, ensuring that the analysis is not biased by the varying volume of content produced by different media outlets.

Across all analyzed websites, SDG-related articles constitute approximately 11.7% of the total content. This global percentage serves as an overall indicator of how sustainability topics are integrated into the editorial agendas of Arab media. However, this figure encapsulates significant variations and disparities among different countries and regions, which reflect diverse socio-political, economic, and cultural factors.

These disparities become more evident when examining the regional differences in SDG coverage between North African and Middle Eastern media outlets. The data presented in Fig. 3 illustrates a notable regional disparity, highlighting a contrast between North African countries and their counterparts in the Middle East. Media outlets in North Africa tend to focus more on SDG topics than those in the Middle East, reflecting distinct socio-political and economic dynamics within these regions.

Figure 3 SDGs coverage per website.

Tunisie Telegraph, representing Tunisia, has the highest proportion of SDG-related content, accounting for nearly 18% of its articles. Elkhabar in Algeria and Hespress in Morocco follow closely, with more than 15% of their content devoted to SDG issues. Similarly, Sahara Media in Mauritania devotes approximately 13% of its articles to sustainable development issues.

The extensive coverage of the SDGs in North African media can be attributed to the region’s ongoing socio-economic challenges, such as high unemployment, environmental stress, and the critical need for sustainable development to ensure long-term stability. These factors likely drive the media’s increased focus on the SDGs, part of a more extensive discussion about economic development and social equity.

Additionally, the active presence of civil society organizations and international partnerships in North Africa may further explain the heightened emphasis on sustainability issues in regional media.

Middle Eastern media outlets such as Aletihad (UAE), Shabiba (Oman), Sabq (Saudi Arabia), Youm7 (Egypt), Marsalqatar (Qatar), and Kuwait News (Kuwait) have lower proportions of SDG-related content, ranging from slightly more than 10% to 5%. Several factors could explain the reduced emphasis on the SDGs. Many countries in this region have more diversified economies, bolstered by substantial revenues from oil and gas, which may reduce the immediate pressure to address sustainability issues compared to those in North Africa. Furthermore, the media in these countries may prioritize coverage of economic growth, technological advancements, and geopolitical issues over sustainability topics. Varying levels of public awareness and governmental commitment to the SDGs could also influence the differences in media focus between North Africa and the Middle East. North African countries, which are often recipients of significant international development aid, may be more inclined to align their media content with global sustainability narratives. In contrast, the wealthier Gulf countries may perceive SDG issues as less pressing, given their financial capacity to address such challenges independently.

Longitudinal coverage analysis

Although the 2030 Agenda was formally adopted in 2015, the analysis began in 2010 to capture early coverage of sustainable development and the transition from the Millennium Development Goals (MDGs) to the SDGs. This broader timeframe provides a more comprehensive picture of the evolving media narratives surrounding development themes.

A longitudinal analysis of the SDGs’ coverage in the Arab media between 2010 and 2024, illustrated in Fig. 4, reveals a complex landscape of fluctuating priorities and influences. In the pre-SDG period, from 2010 to 2014, sustainable development topics received particular attention, probably due to the ongoing MDGs (Sachs, 2012). During this period, there was relatively high coverage, peaking in 2013 at over 16%, perhaps reflecting the intensification of discussions on the post-2015 agenda. Throughout 2011 and 2012, coverage of the Sustainable Development Goals increased significantly, possibly due to the Arab Spring (Jha & Kırşanlı, 2024), which sparked discussions on social justice, economic development, and governance.

Figure 4 SDGs coverage per year.

The transition to the SDG framework in 2014–2015 was accompanied by a temporary drop in coverage, followed by a gradual increase from 2016 to 2018. As countries began implementing the new global goals, media discourse began to include SDG-related topics. Its peak in 2018, at around 13.5%, indicates the Arab media’s strong commitment to the SDGs.

The COVID-19 pandemic introduced a new dynamic in SDG coverage. Although the proportion of articles linked to the SDGs fell slightly between 2020 and 2022, coverage remained relatively high, hovering around 12%. This sustained attention can be attributed to the pandemic’s role in highlighting issues directly related to the SDGs, such as poverty, inequality and access to healthcare. The crisis has highlighted the interdependence of global challenges, potentially reinforcing the importance of the SDG framework in addressing complex systemic issues.

However, from 2022 onwards, a more pronounced downward trend became evident, with coverage falling to around 11% in 2022 and continuing to decline to around 10% in 2024. This sharper downward trajectory in the years following the pandemic could indicate a shift in media focus towards economic recovery and other pressing issues as societies emerge from the immediate health crisis. It may also suggest a possible “SDG fatigue” (Sumner, Hoy & Ortiz-Juarez, 2020; Fallah Shayan et al., 2022; Kakupa & Shayo, 2021) or normalization of sustainability discourse, leading to less explicit coverage of SDG-related topics. Additionally, competing priorities on global and regional agendas may divert media attention away from sustainability issues.

This longitudinal analysis demonstrates that SDG coverage in Arab media has responded to global initiatives, regional events, and global crises. The observed trends reflect the complex interplay between international development agendas, regional socio-political dynamics, and global health crises. Future research could delve deeper into the qualitative aspects of this coverage, investigating how specific SDGs are prioritized at different times and contexts, and exploring the factors behind the recent decline in coverage. Such information would be valuable for understanding the evolving role of sustainability discourse in shaping public consciousness and political priorities in the Arab world.

SDG priorities in Arab news

Global trends

The analysis of coverage distribution across specific SDGs reveals significant variations in how different goals are prioritized in Arab media. As illustrated in Fig. 5, the coverage is notably skewed towards certain SDGs, reflecting the socio-political and economic realities of the region.

Figure 5 Distribution of SDG labels in Arab news articles.

The most covered SDG is SDG 16: “Peace and Justice Strong Institutions,” with approximately 20% of the total SDG-related content. This high level of coverage can be attributed to the ongoing challenges relating to governance, political stability, and social justice in the region. Many Arab countries face issues such as corruption, intransparency, human rights concerns, and conflict (Wafa, 2019; Kamel, 2021). Due to these challenges, media attention tends to focus on the importance of peace, justice, and strong institutions in ensuring the region’s long-term stability and development.

SDG 3: “Good Health and Well-being” follows closely behind and accounts for a significant share of coverage, approximately 15% of the total. This emphasis is understandable given the region’s diverse health challenges (Kronfol, 2012), including the management of communicable and non-communicable diseases, the impact of conflict on health systems, and, more recently, the effects of the COVID-19 pandemic (Nations, 2020).

SDG 5: “Gender Equality” also receives significant coverage, accounting for approximately 12% of all SDG-related stories. Media attention to this goal highlights the growing awareness of gender equality, driven by local movements and international pressure.

SDG 4: “Quality Education”, representing approximately 10% of the coverage, underscores the importance of education in the region’s development discourse. Education is often viewed as a critical enabler of social mobility and economic growth, and media coverage likely reflects the region’s challenges in providing equitable access to quality education. Issues such as high youth unemployment, educational disparities between urban and rural areas, and the need for educational reform are likely contributing to the prominence of this SDG in the media.

In contrast, some SDGs receive relatively minimal attention in the Arab media. For instance, SDG 14: “Life Below Water” and SDG 15: “Life on Land” account for less than 5% of the total coverage. This limited focus may be indicative of the region’s broader socio-economic priorities. Arab countries, particularly those in the Middle East, often prioritize immediate economic and social concerns over environmental sustainability, which may explain the lower media attention devoted to these goals. However, underrepresenting these SDGs could have long-term implications, as environmental sustainability is increasingly recognized as integral to the overall development agenda.

Similarly, SDG 10: “Reduced Inequalities” and SDG 12: “Responsible Consumption and Production” receive limited coverage. The relatively low focus on inequality could reflect a reluctance to address sensitive issues related to wealth distribution, social stratification, and economic disparities within the media. Meanwhile, the minimal attention to responsible consumption and production may be due to the region’s reliance on traditional economic models that prioritize growth over sustainability, particularly in resource-rich countries.

Country-specific SDG priorities in Arab media

The coverage of SDGs in Arab media reveals interesting patterns and priorities that reflect the unique challenges and focus areas of different countries in the region. While there are similarities with the global trend, several notable differences emerge when analyzing the data from the news websites of ten Arab countries, as shown in Fig. B1 in “SDGS Distribution Per Website”.

Globally, the top three SDGs covered are “Peace and Justice Strong Institutions,” “Good Health and Well-being,” and “Gender Equality.” However, some Arab countries show distinct priorities. For instance, the UAE (Aletihad) places an even stronger emphasis on “Peace and Justice Strong Institutions,” with 40.92% coverage compared to the global 23.42%. This heightened focus may reflect the country’s efforts to strengthen governance and institutions amidst regional geopolitical challenges. Additionally, this aligns with the findings of the Bertelsmann Transformation Index (BTI) (Stiftung, 2024), which ranks the UAE first among Gulf countries in governance.

Morocco (Hespress) also shows a high focus on “Peace and Justice Strong Institutions” (38.52%), reflecting ongoing institutional reforms in North Africa since the Arab Spring. Algeria (Elkhabar) stands out with its primary focus on “Good Health and Well-being” ( 28.32%), surpassing the global average of 19.73%. This improvement could be attributed to the country’s ongoing efforts to improve its healthcare system and address public health concerns, particularly during the COVID-19 pandemic (Klouche-Djedid et al., 2021).

Saudi Arabia places a higher emphasis on “Quality Education” ( 13.56% vs. 9.92% globally), aligning with the kingdom’s Vision 2030 plan, which prioritizes education reform and human capital development (Siambi, 2023; Bataeineh & Aga, 2023). Oman (Shabiba) presents a unique case, with “Decent Work and Economic Growth” ranking second ( 13.01% vs. 5.33% globally). This reflects Oman’s focus on economic diversification and job creation for its young population, as evidenced by the newly promulgated Social Protection Law (Royal Decree No. 52/2023) (ILO, 2023), which aims to provide comprehensive coverage for all workers.

The environmental SDGs receive varying levels of attention across the region, with some countries showing significantly greater attention. Kuwait and Qatar significantly emphasize “Climate Action” ( 5.74% and 5.98%, respectively, compared to 1.75% globally). This increased coverage may be linked to the efforts of these oil-rich countries to diversify their economies and mitigate their vulnerability to climate impacts. Qatar (Marsalqatar) also emphasizes “Affordable and Clean Energy” (3.63%), reflecting the country’s investments in renewable energy projects as evidenced by the Qatar National Renewable Energy Strategy (QNRES) (Qatar General Electricity and Water Corporation (Kahramaa), 2024). Oman’s greater emphasis on “Life Below Water” ( 4.62% compared to 1.34% globally) may be attributed to its extensive coastline and the importance of marine resources to its economy.

The coverage of “Zero Hunger” and “No Poverty” SDGs varies significantly across the region. It is particularly prominent in Mauritania (Sahara Media), with 9.03% and 6.46%, respectively, compared to global figures of 4.40% and 3.83%. This heightened focus reflects Mauritania’s ongoing challenges with food security and poverty reduction, which are critical for the country’s development. The severity of these challenges is underscored by the fact that 58.4% of the population live in multidimensional poverty (United Nations Development Programme, 2023). Similarly, Egypt’s Youm7 highlights “Zero Hunger” (6.35%) and “No Poverty” (5.04%), aligning with the nation’s ongoing efforts to tackle poverty and enhance food security. These priorities are crucial in Egypt’s development agenda, as the country faces high poverty rates.

Several countries show significant differences in SDG rankings compared to the global trend. The UAE ranks fifth in “No Poverty” (vs. eighth globally), indicating a stronger focus on social welfare and poverty alleviation. Algeria and Saudi Arabia rank fifth in “Sustainable Cities and Communities” (vs. ninth globally), suggesting an increased emphasis on urban development and sustainability in these rapidly growing economies.

Regional differences:

The analysis reveals distinct patterns in SDG coverage between North Africa and the Middle East, reflecting not only divergent development priorities but also the underlying policy frameworks and national strategies of each region.

In North Africa, countries such as Morocco and Tunisia strongly emphasize governance-related SDGs, particularly “Peace and Justice, Strong Institutions.” This emphasis mirrors ongoing institutional reforms and anti-corruption efforts following the Arab Spring. Morocco’s National Strategy for Sustainable Development (SNDD), adopted in 2017, places governance and social equity at its core, aligning with its high media attention to these themes. Tunisia’s continued democratization process has similarly brought institutional resilience and citizen participation into the national discourse, reflected in its media coverage.

North African countries also consistently address education, health, and gender equality. Mauritania is a notable case, with substantial media attention to hunger and poverty. This attention aligns with its high multidimensional poverty rate and the strategic focus of the 2016-2030 Strategy for Accelerated Growth and Shared Prosperity (SCAPP), which targets food insecurity and rural development. Egypt, facing high poverty levels and rapid population growth, has launched multiple national programs such as “Decent Life” (Hayah Kareema), which aims to improve living conditions in rural areas-an effort that aligns with its heightened media focus on “Zero Hunger” and “No Poverty.”

In contrast, the Middle Eastern sub-region prioritizes economic diversification and environmental sustainability, especially in the Gulf states. This is reflected in the media’s heightened attention to “Decent Work and Economic Growth,” “Climate Action,” and “Affordable and Clean Energy.” These trends align with strategic documents such as Saudi Arabia’s Vision 2030 and Qatar’s National Vision 2030, which emphasize transitioning from oil dependence to knowledge-based economies, innovation, and environmental stewardship. For example, Qatar’s investment in renewable energy, as outlined in its National Renewable Energy Strategy, is reflected in an increased media focus on clean energy initiatives.

Gulf countries also exhibit substantial differences in their approaches to governance-related SDGs. While the UAE gives exceptional media attention to “Peace and Justice Strong Institutions,” suggesting a strong institutional development agenda recognized in governance indices, Saudi Arabia places less emphasis on this SDG in its media, instead prioritizing education and urban development-domains that are central to its Vision 2030 human capital and smart city initiatives.

Oman and Kuwait provide further contrasts. Oman’s focus on “Life Below Water” aligns with its Blue Economy Strategy, which aims to harness its maritime resources for sustainable development. Kuwait’s higher-than-average attention to “Climate Action” and “Sustainable Cities and Communities” reflects its vulnerability to environmental risks and recent policy actions such as the National Adaptation Plan (NAP 2019-2030).

These patterns highlight the impact of national development strategies, economic structures, and recent policy initiatives on SDG media coverage. North African countries tend to emphasize foundational governance and social challenges, while Middle Eastern states, particularly in the Gulf, prioritize futureoriented objectives such as sustainability, economic modernization, and energy transition.

In conclusion, regional SDG media coverage reflects more than thematic preferences-it embodies country-level agendas and developmental milestones. The differences across Arab countries provide valuable insight into how media discourse is shaped by national efforts to achieve the 2030 Agenda.

Crisis impact and gender equality trends

Crisis impact on SDG-related articles: The evolution of SDG coverage in Arab media reveals significant fluctuations that correlate with regional crises. For the “Peace, Justice, and Strong Institutions” SDG, a pronounced increase occurred in 2011 (Fig. 6), coinciding with the onset of the Arab Spring. This period was marked by widespread civil unrest, political upheaval, and demands for greater democratic governance across many Arab countries.

Figure 6 Trends in SDG coverage: Health (SDG 3), Peace (SDG 16), and Gender Equality (SDG 5).

For the “Good Health and Well-being” SDG, the data shows a substantial rise in coverage in 2020 (Fig. 6), clearly reflecting the global impact of the COVID-19 pandemic. The health crisis brought unprecedented attention to public health systems, healthcare accessibility, and overall well-being, with media coverage sharply focusing on these issues during this period (United Nations Development Programme, 2021).

These observable trends in the data not only align with major crises but also serve as further validation of the models used in this study. The data’s alignment with real-world events strengthens the credibility of the analysis, demonstrating the model’s capacity to accurately capture and reflect the region’s sociopolitical dynamics.

Gender equality trends: In terms of “Gender Equality,” two significant increases in media coverage are noted: one in 2012 and another in 2015 (Fig. 6). The 2012 increase aligns with a global focus on women’s rights following the Arab Spring, where gender equality issues were brought to the forefront as part of broader discussions on human rights and social justice (Hussain & Haj-Salem, 2023). The 2015 increase may correspond with the global push towards the SDGs, as this year marked the adoption of the 2030 Agenda for Sustainable Development, which emphasized gender equality as a key priority.

This section highlights the interconnectedness of sociopolitical events and media representation by combining the analysis of crisis impacts with trends in gender equality. It further substantiates the relevance and accuracy of the models applied in this study.

Comparison with UN-Arab region SDG report

The Mohammed bin Rashid School of Government and UN Sustainable Development Solutions Network have produced the Arab Region SDG Index and Dashboards Report 2023/2024 (Mohammed bin Rashid School of Government (MBRSG) & UN Sustainable Development Solutions Network (SDSN), 2024), which measures progress on SDGs in 22 Arab countries. The tool provides aggregate scores for all 17 SDGs for each Arab nation. Figure 7 shows the ranking of the countries based on the index scores reported in the report for all 10 countries covered in this study.

Figure 7 Ranking of covered Arab countries based on the 2023/2024 Arab Region SDG Index Scores.

Country ranking comparison

The report’s regional focus and country-level scoring make it an invaluable resource for comparing media coverage of sustainable development in Arab countries with official benchmarks.

Comparing media interest in the SDGs in Arab countries with the UN rankings highlights both similarities and differences. Tunisia, Morocco, Algeria, Oman, and the United Arab Emirates rank highly in both areas, suggesting a shared commitment to the Sustainable Development Goals. The overlap between media content and government priorities suggests a broader societal agreement, where both the public and authorities view these goals as crucial.

Notable discrepancies emerge when comparing the rankings of Egypt and Mauritania. Egypt ranks relatively high in the Arab SDG Index, reflecting stronger performance across several official indicators. In contrast, Mauritania appears lower in the UN-based ranking but receives greater emphasis in the mediabased analysis. This contrast suggests a divergence between official priorities and the issues highlighted in public discourse. Such misalignments may result from multiple factors, including differences in how national progress is measured vs. how challenges are experienced and discussed by society. The perceived urgency, social visibility, or media framing of particular SDGs may also influence coverage patterns independently of institutional performance.

Mauritania stands out among the cases that illustrate such discrepancies. Despite its lower overall ranking in the Arab SDG Index, the country exhibits substantial media coverage, especially on SDG 1 (No Poverty) and SDG 2 (Zero Hunger). This divergence is not incidental. It reflects the persistent and complex development challenges that the country faces.

Mauritania has long struggled with entrenched poverty, chronic food insecurity, and rural marginalization—issues that remain central to public awareness and national strategies. The government’s Strategy for Accelerated Growth and Shared Prosperity (SCAPP 2019–2030) explicitly identifies these areas as top priorities. However, limited institutional capacity, economic constraints, and implementation difficulties have hindered substantial progress, resulting in low official SDG scores.

In this context, the intensity of media coverage is not necessarily a signal of success but rather a reflection of how pressing and visible these issues are within society. The media’s attention to these goals may serve to spotlight unresolved problems, raise awareness, or express collective concern, especially where institutional mechanisms have struggled to deliver tangible improvements. Thus, the discrepancy between media emphasis and official performance may be interpreted as an indicator of critical unmet needs rather than misalignment per se.

The discrepancy observed in the Mauritania case illustrates a broader phenomenon: media attention and official SDG performance are not always aligned, and their relationship should be interpreted as correlative rather than causative.

A country with a low score on the SDG Index may still receive significant media coverage on development issues precisely because of its inability to address key challenges. The visibility of these challenges in the media often reflects public concern, social tension, or policy gaps, rather than indicating effective action or progress. In such contexts, the media serves as a platform for raising awareness of unresolved or critical problems, thus increasing coverage even as official performance remains low.

Conversely, countries that show alignment between official scores and media attention may do so for different reasons. Media and government priorities may sometimes be synchronized through coherent national strategies, public communication efforts, or shared institutional narratives. Alignment might occur when media environments are tightly controlled, shaping public discourse to reflect official agendas. Although this paper does not assess media freedom or ownership, such structural dimensions may also play a role and warrant further study.

In summary, high media attention should not be interpreted as an indicator of high performance, just as low media coverage does not necessarily imply poor development outcomes. The observed discrepancies underscore the importance of examining local development realities, media dynamics, and institutional capacity when interpreting SDG narratives across the Arab region.

Regional SDG challenges and media coverage

Comparing the official UN regional comparison from the Arab SDG Index report with the study’s regional comparison of SDG coverage in media outlets reveals both alignments and divergences in the portrayal and prioritization of sustainable development challenges across the Arab region.

The UN report and the media study both highlight the significance of SDG 5 (Gender Equality) as a major challenge across the Arab region. The UN report indicates red scores for this goal in both North Africa and GCC countries, mirrored in the media study’s consistently high coverage of gender equality issues across most countries. This alignment suggests a shared recognition of the importance of addressing gender disparities in the region.

Similarly, SDG 16 (Peace, Justice and Strong Institutions) emerges as a key focus in both assessments. The UN report identifies this as a challenge, particularly for North African countries, which aligns with the media study’s findings of high coverage for this goal, especially in North African outlets such as Hespress (Morocco) and Tunisie Telegraph (Tunisia). This correlation suggests a robust public discourse surrounding governance and institutional reform in these countries.

The UN report for both regions highlights economic challenges, as represented by SDG 8 (Decent Work and Economic Growth). The media study reflects this concern, albeit with varying degrees of coverage across different countries. This alignment underscores the region-wide focus on economic development and job creation.

The UN report’s identification of major challenges in SDG 13 (Climate Action) for GCC countries partially aligns with the findings of the media study. Higher coverage of climate action in some Gulf state outlets, such as Kuwait News and Marsalqatar, suggests an increasing awareness of environmental issues in these countries, possibly driven by their vulnerability to climate change and efforts to diversify their economies.

However, there are some discrepancies between the official assessment and media coverage. While the UN report shows better results on SDG 1 (No poverty) for some Gulf countries and greater challenges for North African countries, the media study reveals relatively low coverage of poverty issues in most media outlets, regardless of region. There may be a disconnect between official development priorities and media attention to this issue.

Interestingly, the media study reveals strong coverage of SDGs 3 (Good Health and Well-Being) and 4 (Quality Education) in both regions. However, they are not considered significant regional challenges in the UN report. This increased media attention may reflect public interest in these fundamental aspects of human development, even if they are not the most critical challenges, according to official assessments.

The UN report notes challenges related to SDG 9 (Industry, Innovation and Infrastructure) in both regions, but this goal receives relatively little coverage in most of the media outlets studied. This divergence may suggest the need for increased public awareness and discourse on innovation and infrastructure development.

The media study provides a nuanced analysis of regional challenges in comparison to the findings of the UN report. African media coverage highlights governance and social concerns, whereas Gulf media concentrates on economic diversification and environmental sustainability. While not clearly defined in the UN report’s regional comparison, these distinctions may reflect the development trajectories and priorities of each subregion.

In summary, the analysis of the UN evaluation and the examination of media reports show similarities and differences in how sustainable development issues in the region are portrayed. The fact that the media extensively covers the challenges mentioned in the UN document indicates a synchronization between agendas and public discussions. Nevertheless, discrepancies between the two highlight instances where public understanding and official evaluations may not align completely, revealing opportunities for improved communication and involvement in specific sustainable development objectives. The examination highlights the interplay between established development goals and media portrayals that influence discussions on sustainable development in the Arab region.

Table 4 shows a comparison between the SDG coverage in the media and official coverage in the UN SDG report. We defined alignment as the overlap between the top five SDGs covered in the media and the top five SDGs with the highest performance or priority based on the UN SDG Index. One can observe that there is a strong alignment in countries such as the UAE, where media emphasis on goals like education (SDG 4), health (SDG 3), and poverty reduction (SDG 1) closely mirrors national SDG performance. In contrast, Egypt and Morocco exhibit misalignment; for instance, while SDG 13 (Climate Action) shows notable progress in both countries according to the SDG Index, it receives minimal media attention, indicating a disconnect between policy achievements and media narratives. Saudi Arabia shows partial alignment, with SDG 3 being prioritized both in media and official reporting.

Table 4 Media-SDG coverage vs. Official performance per Country.

Country	Top 5 SDGs in media coverage	Top 5 SDGs in official progress reports	Alignment (Yes/No)	Comments	
Saudi Arabia	SDG 3, 4, 5, 16, 11	SDG 1, 3, 9, 10, 17	✓ (1/5)	Partial alignment; overlaps with SDG 3	
Morocco	SDG 16, 3, 5, 4, 8	SDG 13, 1, 12, 11, 7	X (0/5)	Under-representation of environmental goals in media	
Egypt	SDG 16, 3, 5, 11, 2	SDG 13, 1, 10, 12, 7	✓ (0/5)	Misalignment; media focus differs from official performance	
UAE	SDG 16, 5, 3, 4, 1	SDG 1, 10, 9, 3, 4	✓ (3/5)	Good mediapolicy alignment	

Several factors can contribute to the observed alignment and divergence between government priorities and media coverage. Geopolitical interests and national policies often drive governments to focus on specific SDGs that align with their broader national agendas. For instance, Gulf countries may prioritize.

SDGs related to economic diversification and climate change, while North African countries may focus more on social issues, such as poverty reduction and gender equality. Crises, such as wars, political instability, or natural disasters, may lead to a temporary shift in focus toward specific SDGs, including health (SDG 3) or economic growth (SDG 8). Similarly, cultural and religious contexts influence how SDGs, such as gender equality or peace and justice, are framed and discussed in the media. Public opinion, shaped by civil society movements and grassroots campaigns, can also significantly influence which SDGs receive media attention. Moreover, the media ownership structure and economic factors influence the topics that media outlets prioritize, often reflecting the interests of owners or advertisers, which may not always align with government priorities. Media freedom and government influence are also factors worth exploring in future studies, as they shape media coverage of certain SDGs through direct or indirect pressures on content.

In conclusion, while a strong correlation exists between government priorities and media coverage of SDGs in the Arab region, this relationship is influenced by various factors, including geopolitical, economic, cultural, and societal dynamics. The observed alignment should not be interpreted as a direct causal effect, but rather as a correlation shaped by complex interactions among governments, media, and public opinion. The discrepancies in some areas underscore the potential for further investigation into how different media outlets and governments prioritize and frame SDG-related issues, as well as how these dynamics may influence public discourse and policy decisions.

To better align media discourse with sustainable development objectives, policymakers should consider partnering with media outlets to promote underrepresented SDGs through targeted awareness campaigns and incentives for balanced reporting. The AI-driven models developed in this study offer scalable tools for governments and NGOs to monitor media trends, detect thematic gaps, and adjust policy communication strategies accordingly. Media organizations, in turn, can utilize these tools to audit their coverage in real-time, identify potential biases, and ensure a more comprehensive representation of the 17 SDGs. Incorporating such AI systems into newsroom workflows can also support editorial planning, training programs, and diversity audits, enhancing the quality and equity of public discourse around sustainability.

Conclusion

In conclusion, this study examined the extent to which the Sustainable Development Goals are covered in Arab media, with a focus on their alignment with official priorities. A comprehensive dataset of over 1.2 million Arabic news articles spanning over a decade was developed to assess how Arab media reflects public and governmental interest in the SDGs. To analyze this data, we employed a novel approach that combines deep learning, large language models, and synthetic data generation methods rarely applied in Arabic-language media research. This work’s scope, linguistic focus, and methodological integration surpass those of previous efforts in the literature.

The analysis revealed notable regional differences in SDG coverage, with North African media exhibiting higher coverage rates than those in the Middle East. These differences highlight distinct socio-political and economic dynamics, such as the higher emphasis on governance and institutional reform in North Africa. Additionally, while media in the Gulf states tend to focus more on economic development and environmental sustainability, SDGs related to social issues, such as gender equality and education, received considerable attention across the region.

The alignment between media coverage and official government priorities was generally strong, although some discrepancies were noted. For example, countries such as Tunisia, Morocco, and Algeria closely aligned with governmental priorities. In contrast, others, such as Mauritania, demonstrated a media focus on issues like hunger and poverty, which might differ from official priorities. These insights into the media’s representation of the SDGs provide valuable perspectives for policymakers and researchers, contributing to a deeper understanding of how public discourse aligns with the Sustainable Development Goals in the Arab world.

Moreover, our findings offer several practical applications for key stakeholders. For policymakers, understanding the alignment (or misalignment) between media discourse and official SDG priorities enables more targeted communication strategies and better-informed public policy. For example, countries where key SDGs such as climate action are underrepresented in the media may consider public awareness campaigns to fill the gap. Media professionals can also use these insights to reflect on their editorial focus and enhance coverage of neglected yet nationally important SDGs, thereby fostering more balanced reporting. Lastly, international development organizations and NGOs working in the Arab world can use the country-specific trends identified in this study to guide localized interventions and partnerships. This study also demonstrates the utility of AI tools for large-scale media analysis in low-resource languages, which could be adopted by government agencies, research institutes, or civil society groups.

Appendix

A ChatGPT prompt

Produce Arabic language text in the style of a news article. Ensure that no English words are included.

As an expert in Sustainable Development Goals (SDGs) Arabic articles generation, you possess the capability to discern whether a text covers any of the UN SDGs or not. Your expertise extends to generating both SDG-related Arabic text and non-SDG-related text.

Generate relatively short texts composed of four separate paragraphs, maintaining a formal and clear style. When prompted with \SDG{X}, produce a text related to the SDG number X.

Here are two examples of how the number of the SDG is indicated: \SDG {1} : means you should generate a text related to SDG 1

\SDG {10} : means you should generate a text related to SDG 10.

Upon receiving the \NonSDG command, generate a general text composed of four separate paragraphs that avoids covering any of the SDGs.

The SDG-related text should not merely provide a definition or introduction but rather adopt a news article style to discuss events, analyze problems, or present ideas related to the SDG. It must not explicitly mention the SDG name or number.

To differentiate between SDG-related and non-SDG-related news articles, consider the following criteria: Topic focus:

SDG-related article: Directly addresses one or more of the United Nations Sustainable Development Goals, discussing initiatives, progress, challenges, or successes related to achieving specific SDGs.

Non-SDG-related article: Covers topics unrelated to the SDGs, such as general news, events, discoveries, trends, or other subjects not directly tied to sustainable development goals. Content emphasis:

SDG-related article: Features discussions or actions related to sustainability, social progress, environmental protection, economic development, or other themes aligned with the SDGs. May include references to specific SDGs, targets, or indicators.

Non-SDG-related article: Covers a wide range of topics, including politics, entertainment, science, technology, sports, or human interest stories, without directly focusing on sustainable development goals. Language and terminology:

SDG-related article: Uses terms and language associated with sustainable development, such as “sustainability,” “inclusive growth,” “climate action,” “gender equality,” or “poverty alleviation.” May explicitly mention the SDGs or refer to related international agreements, frameworks, or organizations.

Non-SDG-related article: Does not include sustainability-specific terms or references to international development goals. Uses language appropriate to the subject matter discussed. Purpose and perspective:

SDG-related article: Aims to inform, raise awareness, advocate, or report on efforts to address global challenges and advance sustainable development. Highlights the importance of collaboration, innovation, and collective action in achieving the SDGs.

Non-SDG-related article: Aims to inform, entertain, or engage readers without a specific focus on sustainable development goals or related issues.

Below are some examples of SDG-related texts along with the corresponding SDG number:

SDG: 6

SDG: 16

SDG: 8

SDGS distribution per website

Figure B1 Distribution of SDG labels in Arab news articles per website.

Supplemental Information

Supplemental Information 1 Binary and Multiclassification Training Code.

Additional Information and Declarations

Competing Interests

The authors declare that they have no competing interests.

Author Contributions

Mohammed Alsuhaibani conceived and designed the experiments, analyzed the data, authored or reviewed drafts of the article, and approved the final draft.

Kamel Gaanoun conceived and designed the experiments, performed the experiments, analyzed the data, performed the computation work, prepared figures and/or tables, authored or reviewed drafts of the article, and approved the final draft.

Ali Mustafa Qamar analyzed the data, authored or reviewed drafts of the article, and approved the final draft.

Data Availability

The following information was supplied regarding data availability:

The fully translated dataset is available at Zenodo: Gaanoun, K., & Alsuhaibani, M. (2025). Arabic OSDG Dataset [Data set]. Zenodo. https://doi.org/10.5281/zenodo.15257003.

The synthetic dataset is available at Zenodo: Gaanoun, K., & Alsuhaibani, M. (2025). Arabic Sustainable Development Goals (SDG) Synthetic News dataset (Version v1) [Data set]. Zenodo. https://doi.org/10.5281/zenodo.15255498.

The code is available in the Supplemental files.

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
