# Peer review of "Artificial intelligence-driven insights into Arab media’s sustainable development goals coverage"

_PeerJ Computer Science, doi:10.7717/peerj-cs.3071_

## Round 0.1 · original submission · Major Revisions

Two reviewers have provided detailed feedback. Please revise your manuscript accordingly.

·

Basic reporting

The article builds on the media coverage importance for achieving the SDGs by analyzing a large dataset of Arab news articles. The introduction is well structured, and the technical contributions are highlighted. Nonetheless, it does not clear inform how the “correlation” between media coverage and official SDG priorities will be assessed. I would also advise the authors to emphasize the non-English marginalization process in this research area, for me one of the greatest contributions of the paper. In literature review, I’d like to see a table pointing out the differences between your study and previous research.
The research question is well defined, although the authors could emphasize the regional analysis by country, as it contributes to expand the knowledge regarding SDG actions.
The language is adequate, and the paper storytelling is easy to follow.

Experimental design

Most of the methodological details are detailed in section 4. The authors have made an effort to detail the choices and to be as transparent as possible regarding the data augmentation. Although the balance between SDG related and non-related articles was not achieved, the dataset seems safe to support the model training.
A high technical standard was considered, by using frontier models for training. The use of ChatGPT for data augmentation might cause some discussion. Although the prompt is transparent and allows replication of the process, and the data is available, the use of an American-centered LLM might have biased the results. Nonetheless, all methods have strengths and weaknesses, and in order to make more clear the authors consciousness, a deeper comment about this could be presented in the conclusion section.

Validity of the findings

The findings are supported by the data provided. As a minor, Figure 6 presents the trends in SDG coverage from 2010 onwards. However, the 2030 Agenda was only released in 2015, so I believe that, if the authors would choose to keep this analysis, a comment about this timelapse could benefit the paper.
The data have been provided, as well as the code.

Additional comments

I thank the opportunity to get to know such an interesting research, and congratulate the authors for their efforts.

·

Basic reporting

The study presents a compelling analysis of Arab media’s coverage of the UN Sustainable Development Goals (SDGs) using artificial intelligence, particularly deep learning and synthetic data augmentation. The research is timely, methodologically innovative, and provides valuable regional insights. However, there are areas where improvements can be made to enhance clarity, methodological transparency, and the depth of discussion.
Before accepting for publication, the following points should be implemented.
1. The manuscript should better address the practical applications of its findings, ensuring accessibility for a broader readership.
2. Manuscripts submitted to the journal are expected to be written in good English. Several sentences are unclear and grammatically incorrect (e.g., "An experimental program was conducted carried out"). Phrasing such as "and gives very satisfactory resistance in the long term" should be refined to enhance clarity. A thorough language revision is needed to improve readability.
3. The abstract should explicitly answer the following questions: a) What issue was the focus of the study, and why is it significant? b) What methods were used? c) What are the important results? d) What inferences can be made based on the findings? e) What is the novelty of the work, and how does it go beyond previous efforts in the literature?
4. The Abstract and Conclusions sections should also include brief, clear statements about the novelty of the work.
5. The introduction should make a compelling case for the study's usefulness and novelty by addressing: a) What is already known in the open literature? b) What are the gaps in research that need to be addressed? c) What needs to be done, why, and how?
6. The authors should provide a more detailed context for their study by outlining the existing challenges or gaps in current research that their work aims to address in introduction section.
7. It is important to compare and discuss the results and findings with earlier work in the literature. Kindly add a comparative discussion.
8. The paper introduces deep learning models and synthetic data augmentation for analyzing SDG-related content. However, the specifics of model architecture, training data, validation techniques, and evaluation metrics are not detailed.
9. More information is needed on how Transformer-based models were implemented. Were pre-trained models fine-tuned, or were they trained from scratch?
10. A comparative analysis of model performance with and without synthetic data augmentation would strengthen the justification for using this technique.
11. The study identifies differences in SDG coverage between North Africa, the Gulf States, and the Middle East, but it does not provide a deeper discussion of why these differences exist.
12. The Mauritania case is briefly mentioned as an outlier but lacks an explanation. A deeper analysis of why certain nations diverge from official priorities would improve the discussion.
13. The study finds a strong correlation between media coverage and government priorities but does not fully explain the mechanisms driving this alignment.
14. Are governments actively influencing media discourse, or does media organically reflect national priorities? Including a discussion on media freedom and government influence would strengthen the argument.
15. While the study mentions policy implications, concrete recommendations are missing.
16. How can policymakers leverage AI-driven insights to enhance media engagement with SDGs?
17. Could media organizations use AI models to track biases or improve reporting balance across SDG themes?

Experimental design

As above

Validity of the findings

No validation of result have not been found.

---

## Round 0.2 · Minor Revisions

This version has undergone significant improvements. Please strengthen the validation as required.

·

Basic reporting

no comment

Experimental design

no comment

Validity of the findings

In general, conclusions are well supported. However, the analysis of “alignment with official priorities” is somewhat vague. While country comparisons are made, there is limited quantitative grounding in actual government SDG policies or reports. I think the paper will benefit from specific examples or a table showing mismatches between media coverage and country-level SDG performance. I believe that this would help to validate the alignment/misalignment claims.

Additional comments

no comment

---

## Round 0.3 · accepted · Accept

Many thanks to the authors for their efforts to improve the work. This version satisfied the reviewers successfully. I believe it can be accepted now. Congrats!

·

Basic reporting

Clearly written.

Experimental design

No experiment has been conducted.

Validity of the findings

Acceptable

Additional comments

The manuscript could be accepted for final publication.

·

Basic reporting

No Comment

Experimental design

No Comment

Validity of the findings

No Comment

Additional comments

No comments